# A Late Cretaceous amber biota from central Myanmar

Daran Zheng [1,2], Su-Chin Chang[2], Vincent Perrichot [3], Suryendu Dutta[4], Arka Rudra[4], Lin Mu[1], Ulysses Thomson[1,5], Sha Li[1], Qi Zhang[1], Qingqing Zhang[1], Jean Wong[2], Jun Wang [2], He Wang[1], Yan Fang[1], Haichun Zhang[1] & Bo Wang [1,6]

Insect faunas are extremely rare near the latest Cretaceous with a 24-million-year gap spanning from the early Campanian to the early Eocene. Here, we report a unique amber biota from the Upper Cretaceous (uppermost Campanian ~72.1 Ma) of Tilin, central Myanmar. The chemical composition of Tilin amber suggests a tree source among conifers, indicating that gymnosperms were still abundant in the latest Campanian equatorial forests. Eight orders and 12 families of insects have been found in Tilin amber so far, making it the latest known diverse insect assemblage in the Mesozoic. The presence of ants of the extant subfamilies Dolichoderinae and Ponerinae supports that tropical forests were the cradle for the diversification of crown-group ants, and suggests that the turnover from stem groups to crown groups had already begun at ~72.1 Ma. Tilin amber biota fills a critical insect faunal gap and provides a rare insight into the latest Campanian forest ecosystem.

[1] State Key Laboratory of Palaeobiology and Stratigraphy, Nanjing Institute of Geology and Palaeontology and Center for Excellence in Life and Paleoenvironment, Chinese Academy of Sciences, 39 East Beijing Road, 210008 Nanjing, China. [2] Department of Earth Sciences, The University of Hong Kong, Hong Kong Special Administrative Region, Hong Kong, China. [3] University of Rennes, CNRS, Géosciences Rennes - UMR 6118, 35000 Rennes, France. [4] Department of Earth Sciences, Indian Institute of Technology Bombay, Mumbai 400076, India. [5] School of Earth Sciences, University of Bristol, Life Sciences Building, Tyndall Ave., Bristol BS8 1TQ, UK. [6] Shandong Provincial Key Laboratory of Depositional Mineralization & Sedimentary Minerals, Shandong University of Science and Technology, 266590 Qingdao, Shandong, China. Correspondence and requests for materials should be addressed to S.-C.C. (email: suchin@hku.hk) or to B.W. (email: bowang@nigpas.ac.cn)

Angiosperm-dominated forests became worldwide prior to the latest Cretaceous (Maastrichtian) and the mass extinction at the Cretaceous-Paleogene (K-Pg) boundary[1–3]. Despite the large number of Cretaceous and Cenozoic insect faunas to date, there is a 24-million-year gap spanning from the early Campanian to the early Eocene, which markedly hinders our understanding of the reorganization of terrestrial ecosystem and the impact of the K-Pg extinction event on the evolution of insects[4–6].

The present study reports a unique Tilin amber biota from central Myanmar, which provides new insight into the vanished tropical forest during the latest Campanian.

## Results

**Studied material.** The studied amber was collected from the Cretaceous Kabaw Formation of Tilin (21° 41′ N, 94° 5′ E), Gangaw district, Magway region of central Myanmar (Fig. 1; Supplementary Figures 1–4). Tilin is located on the West Burma block which was part of Southeast Asia by the early Mesozoic[7,8]. It was quite near to the Kachin area, where Kachin amber (also called Burmese amber) is unearthed (Supplementary Figures 1, 2). Tilin and Kachin were near the equator during Late Cretaceous, at almost the same locations as the present[8] (Supplementary Figure 1). Pieces of Tilin amber, known for their reddish or yellowish color, are usually smaller than 10 cm (Figs. 2, 3; Supplementary Figure 4d).

**U–Pb geochronology.** One tuff sample (M-1) was collected just above the amber-bearing layers for LA-(MC)-ICP-MS U–Pb

dating (Fig. 1). Overall, 100–300 μm size fractions of zircons from this sample are abundant. Euhedral morphologies and oscillatory zoning patterns of these zircon grains indicate an igneous origin (Supplementary Figure 5). Non-angular grain facets also suggest this tuff deposited near the magmatic source of the zircons. A total of 15 zircons (concordance >98%) were analyzed (Supplementary Figure 5a). Ten analyses provide the youngest age at $72.1 \pm 0.3$ Ma in weighted mean value (MSWD = 0.6; uncertainties are given at the $2\sigma$ level; Supplementary Figure 5b), presenting the maximum depositional age for M-1. Five older ages (73 Ma, 74.4 Ma, 74.9 Ma, 75.6 Ma, and 522 Ma) are considered to be xenocrysts and excluded for calculating the youngest age. The age of tuff, $72.1 \pm 0.3$ Ma, is near the Campanian-Maastrichtian boundary ($72.1 \pm 0.2$ Ma)[9], and indicates a latest Campanian age for the underlying amber-bearing layers.

**Ammonites and their age implications.** Some ammonites were also found preserved in nodules of brown sandstone underlying the amber layer (Fig. 1b). They belong to *Sphenodiscus* sp. (Supplementary Figure 6) based on the following characters: weak ventrolateral tubercles, all saddles of suture normally indented, some auxiliaries entire, first lateral saddle with two distinct adventive lobes and outer one bigger, folioles with long, narrow necks. They cannot be attributed to a species due to the lack of well-preserved characters (see Supplementary Note 1). Fossils from Africa, Europe, and the Americas indicate Sphenodiscid ammonites were widespread during the Maastrichtian. Sphenodiscidae probably originated in the African epicontinental seas during Campanian, with the oldest *Sphenodiscus* associated with *Libycoceras* in the Middle

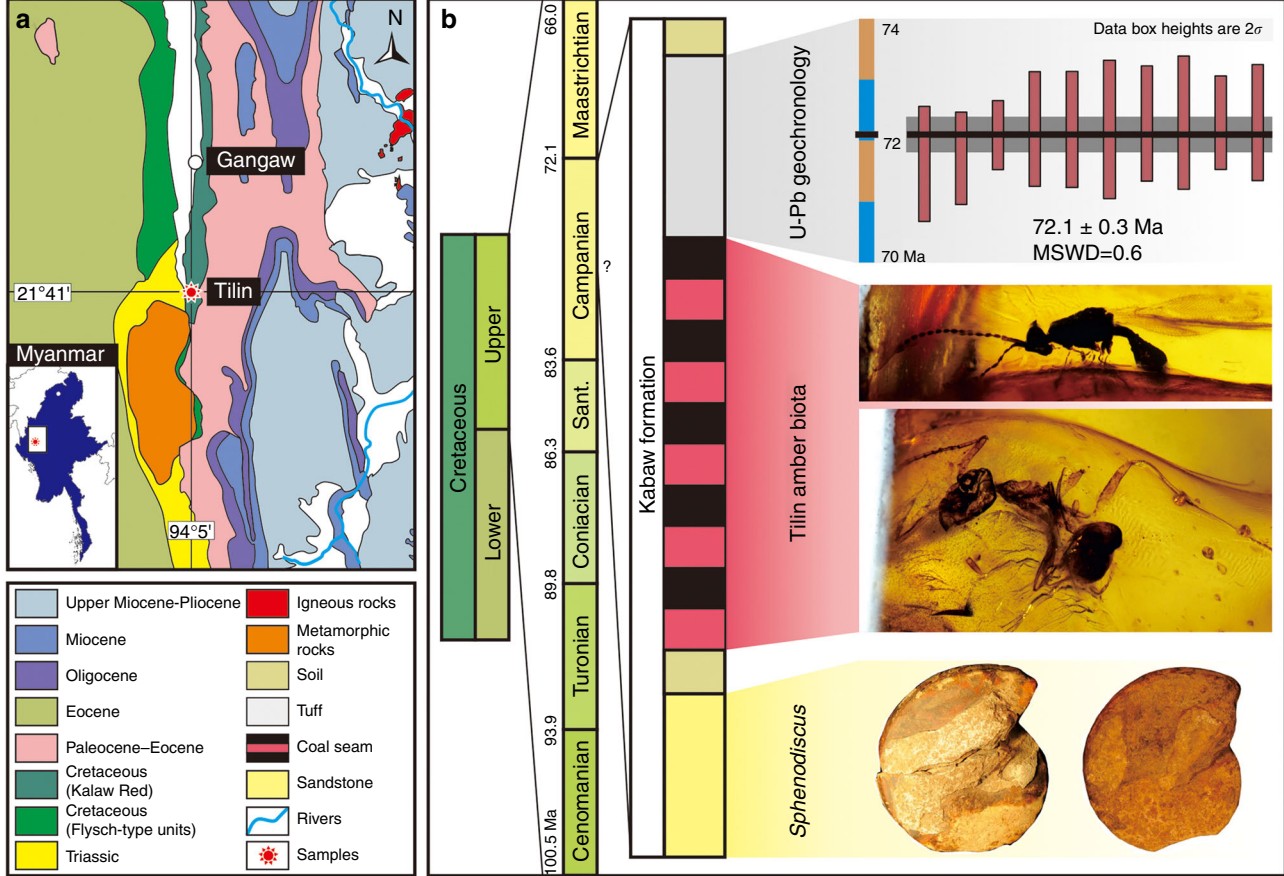

**Fig. 1** Geology of Tilin amber. **a** Geological location of Tilin amber in Gangaw, central Myanmar. **b** Stratigraphic column showing lithologies, sample points, and age results

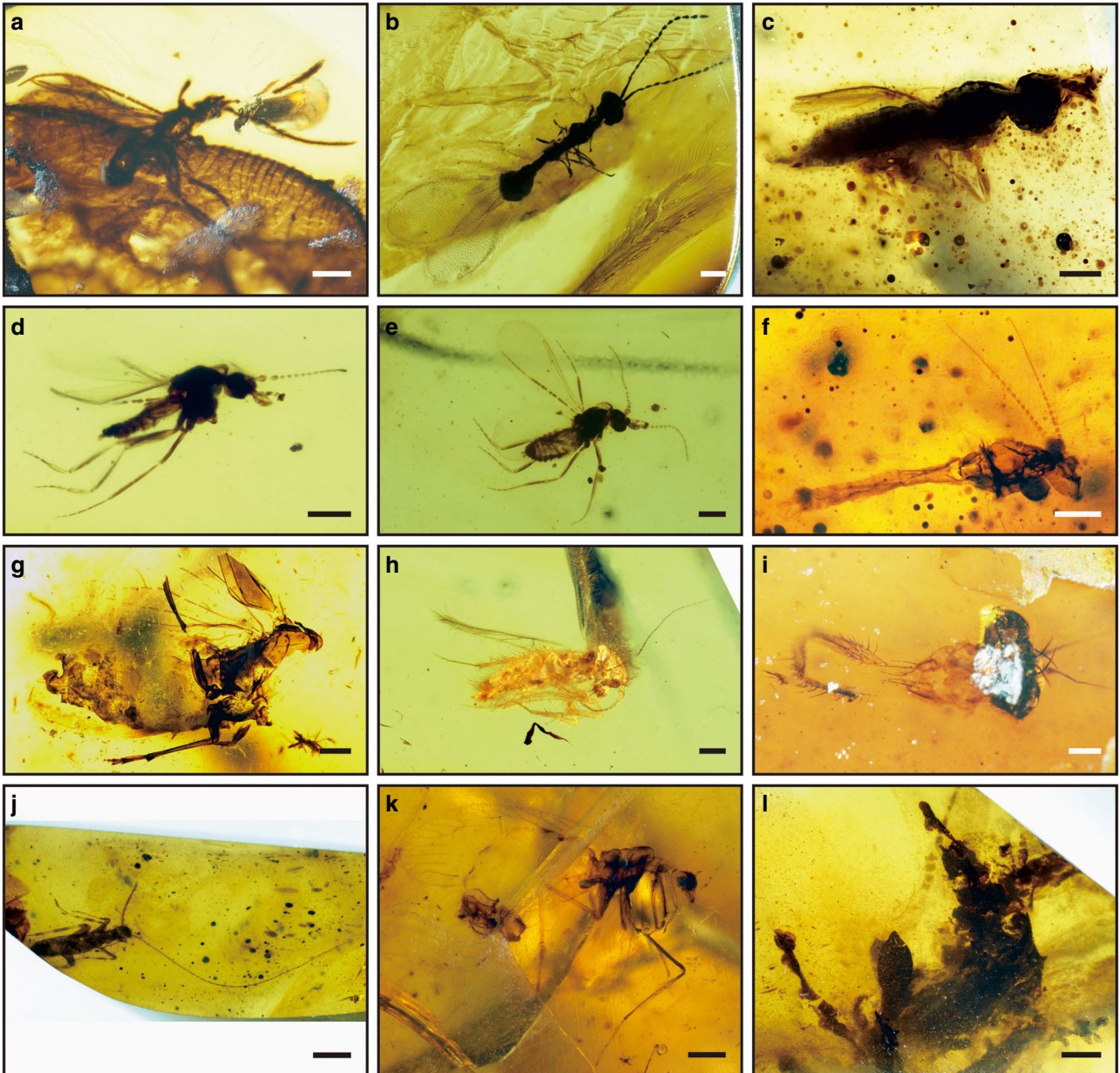

**Fig. 2** Photographs of typical inclusions in Tilin amber. **a–c** Parasitic wasp (Hymenoptera). **a** Braconidae, NIGP168500. **b** Diapriidae, NIGP168501. **c** Scelionidae, NIGP168502. **d**, **e** Biting midges (Diptera: Ceratopogonidae: *Protoculicoides*), NIGP168503. **f** Non-biting midge (Diptera: Chironomidae), NIGP168504. **g** Planthopper (Hemiptera: Achilidae), NIGP168505. **h** Scaly-winged barklice (Psocoptera: Lepidopsocidae), NIGP168506. **i** Cockroach (Blattaria), NIGP168507. **j** Mantis (Mantodea: *Burmantis*), NIGP168508. **k** Beaded lacewing (Neuroptera: Berothidae), NIGP168505. **l** Moss (Bryophyta), NIGP168509. Scale bars, 0.5 mm (**a**, **g**, **k**), 0.2 mm (**b–f**, **h**, **i**), 1 mm (**j**), and 2 mm (**l**)

East, Nigeria, and Peru, indicating a late Campanian appearance[10]. *Sphenodiscus* was widely considered to be restricted to the Maastrichtian, and has occurred in the lower Maastrichtian of Madagascar, but was predominant in the upper Maastrichtian of Europe[11]. *Sphenodiscus siva* was originally recorded from the Maastrichtian, and also reported from the Maastrichtian of Netherlands where it constitutes the oldest *Sphenodiscus* species in Europe[12]. The first appearance of *Sphenodiscus* sp. in the central Myanmar suggests a late Campanian to Maastrichtian age, providing a lower constraint for Tilin amber. Together with the radioisotopic age of 72.1 ± 0.3 Ma for the upper constraint, Tilin amber should be within the latest Campanian age. As such, Tilin amber is at least 27 million years younger than the well-known Kachin amber, which was considered as earliest Cenomanian (98.8 ± 0.6

Ma) in age based on a SIMS U–Pb dating of zircon grains from the amber-bearing stratum[13], and slightly younger based on fossil estimates: the associated palynomorphs suggested a late Albian–early Cenomanian age, and the ammonite *Mortoniceras* indicated a middle Albian–late Albian age[14].

**Pyrolysis gas chromatography mass spectrometry.** The Tilin amber fragments were analyzed using Pyrolysis Gas Chromatography Mass Spectrometry (Py-GC-MS; Fig. 4; Supplementary Table 2). Tilin amber is diagenetically altered, with diagnostic biomarkers degraded. The major pyrolysis products are aromatic compounds such as alkylated benzenes, alkylated naphthalenes and alkylated phenanthrenes (Supplementary Figure 7). Aromatic

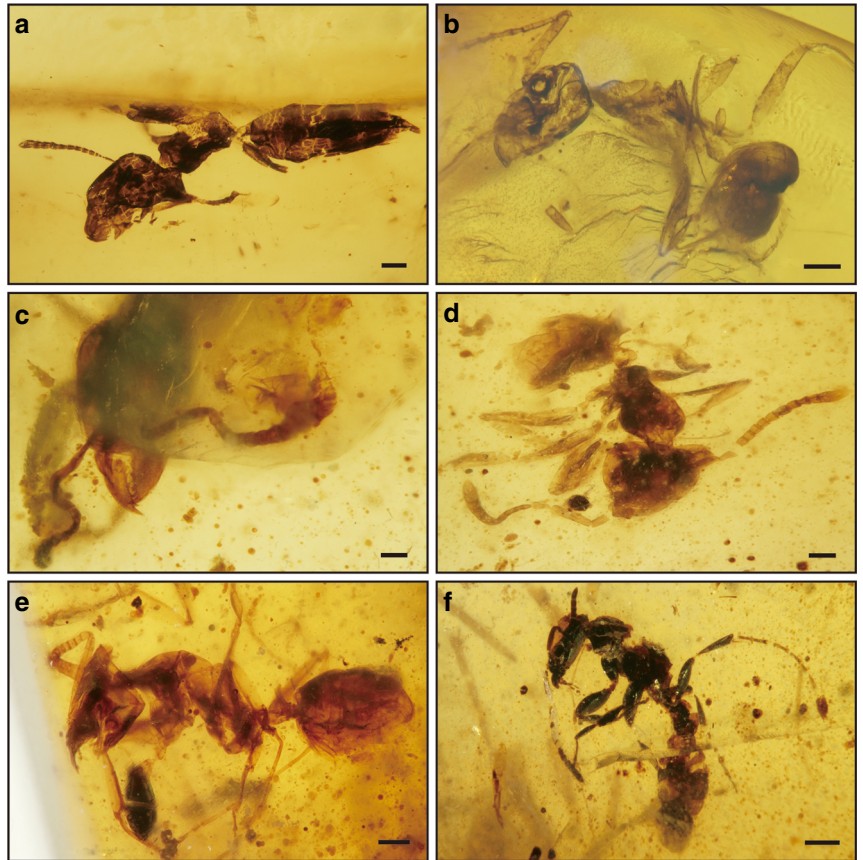

**Fig. 3** Photographs of Formicidae in Tilin amber. **a** Dolichoderinae, genus A, NIGP168510. **b** Dolichoderinae, genus B, NIGP168511. **c** Dolichoderinae, genus C, NIGP168512. **d**, **e** Dolichoderinae, genus B?, NIGP168513. **f**, Ponerinae?, female specimen, NIGP168514. Scale bars, 0.2 mm (**a**, **b**, **d**, **e**), 0.1 mm (**c**), and 0.4 mm (**f**)

diterpenoids were detected such as 16, 17, 18-trisnorabieta-8, 11, 13-triene, bisnordehydroabietane, simonellite, bisnorsimonellite (Supplementary Figure 7), clearly indicating a gymnosperm origin[15]. Ionene and methyl ionene are diagenetic products of sesquiterpenoids, and are present in high abundance. Although Tilin was close to Kachin in West Burma block during Cretaceous (Supplementary Figure 1), Tilin amber is chemically clearly distinct from Kachin amber, which supposedly originated from araucarian or pinaceous trees[15,16]. Angiosperms explosively diversified in mid-Cretaceous time, and became dominant in forests worldwide by the Maastrichtian[1,17]. However, the gymnosperm-derived Tilin amber suggests that gymnosperms were still abundant in the latest Campanian equatorial forests. Nowadays, the forests of Southeast Asia are dominated by dipterocarps (Dipterocarpaceae, a family of angiosperm), and dipterocarp fossils and resins are present in India and Southeast Asia by the Eocene[18]. As such, the replacement of gymnosperms by dipterocarps in Southeast Asian forests most likely occurred from the Maastrichtian to Paleocene.

## Discussion

A total of 52 arthropod and plant inclusions were found in 5 kg of amber (Figs. 2 and 3). These specimens are commonly, more or less, heavily distorted, often partially preserved, with obvious alteration due to dehydration. Among them, 34 insects can be identified to order level, encompassing a notable diversity of 12 families in orders Hymenoptera, Diptera, Hemiptera, Psocoptera, Coleoptera, Blattaria, Mantodea, and Neuroptera. The insect fauna is dominated by Hymenoptera and Diptera (80% of all

insects), as in many other amber biota[4,19]. This dominance relates to a probable dominance of these orders in the forest insect palaeocoenosis and a bias of attraction of these insect groups to the resin. Plant remains assignable to some undetermined bryophytes were also found preserved in amber (Fig. 2l). Most of the insect families in Tilin amber are common elements in Cretaceous and Cenozoic ambers, like Braconidae, Scelionidae, Ceratopogonidae, Chironomidae, and Lepidopsocidae. One exception may be the Diapriidae, which is usually rare in other Cretaceous ambers[4]. All of the identified insect families are also recorded earlier in the Mesozoic, including from Kachin amber. Interestingly, some fossils represent the latest known occurrence of typically Cretaceous taxa, demonstrating the survival of these lineages to the latest Campanian: the biting midge (Ceratopogonidae) genus *Protoculicoides* was known from ambers[20] in Albian (Spain), Cenomanian (France and Kachin), Turonian (New Jersey, USA), Coniacian-Santonian (Taimyr, Russia), and early/middle Campanian (Canada); in contrast, the mantis (Mantodea) genus *Burmantis* is only known from the Barremian and Cenomanian ambers (Lebanese and Kachin)[21].

The new latest Campanian records also comprise lineages of more modern appearance, demonstrating a pre-Cenozoic transition from extinct, stem groups to extant, crown groups. A notable discovery is six worker ants belonging to three unknown genera of the extant subfamily Dolichoderinae, as well as a dealate female likely of the Ponerinae (Fig. 3). Ants are among the most successful groups of insects and can comprise up to 15–20% of the animal biomass in tropical forests[22]. The earliest definitive ants are from mid-Cretaceous French and Kachin ambers[22]. The ants from Tilin amber are one of the earliest records of crown-

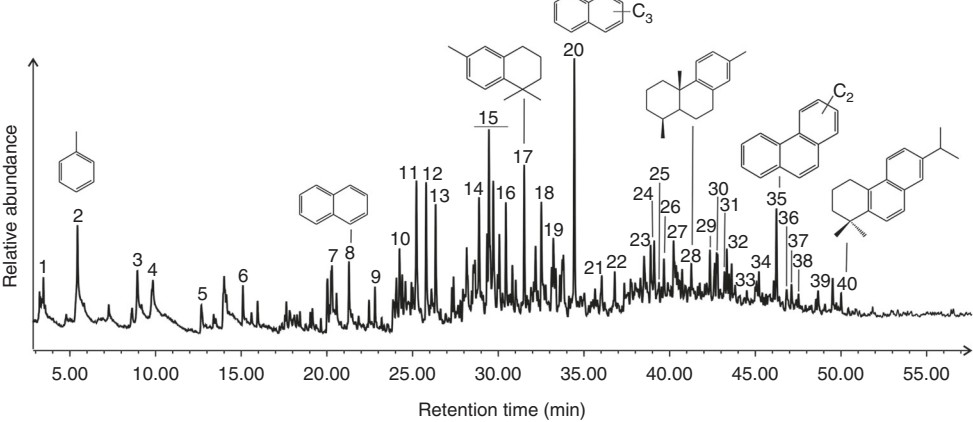

**Fig. 4** Total ion chromatogram of Tilin amber from Py-GC-MS analysis. Identification of numbered peaks is listed in Supplementary Table 2

group ants, as only two Cretaceous species have been definitively assigned to extant subfamilies until now: one formicine from New Jersey Turonian amber and one dolichoderine from Canadian Campanian amber[23,24]. Compression fossils from the Turonian of Botswana were also tentatively assigned to Ponerinae, but the exact placement remains equivocal[25]. All other ants recorded earlier in the Cretaceous belong to extinct subfamilies, mostly Sphecomyrminae[22,25], and all Cenozoic ants are attributable to modern subfamilies, with the exception of the Eocene lineage Formiciinae. After a 24-million-year gap spanning from the early Campanian to the early Eocene, there is a marked change in the composition and number of ants in fossil deposits[25]. The seven ants in Tilin amber reduce this gap and increase the total number of known Cretaceous crown-group ants from 3 to 10. The proportion of ants among total insects is surprisingly high in Tilin amber (20%) compared to other Cretaceous deposits (typically below 2%)[22], but more Tilin amber samples are required for a more accurate estimate. Our findings provide reliable evidence for the Late Cretaceous radiation of crown-group ants, and the apparent absence of sphecomyrmines suggests that the turnover from stem groups to crown groups had already begun by the latest Campanian. The origin and diversification of angiosperms led to marked changes in terrestrial ecosystems during the Late Cretaceous[26], which probably provided a basis for a radiation of ants. In addition, the composition and number of ants in Tilin amber supports that tropical forests were the cradle for crown-group ants.

Tilin amber biota yields the latest known diverse insect assemblage in the Mesozoic[27], and provides a unique window into a vanished tropical forest. The age, chemical components, and inclusions of Tilin amber are different from those of Kachin amber in northern Myanmar, showing a biotic change from mid-Cretaceous to Late Cretaceous. Following ongoing excavation of Tilin amber, future discovery of arthropods and plants are promising and will provide new insight into the coevolution between early angiosperms and insects during the latest Campanian.

## Methods

**Tuff and zircon dating**. The tuff sample was crushed, sieved, and washed first. Zircons were selected using magnetic and heavy liquid separations. A total of 80–200 μm inclusion-free zircon grains from the sample were hand-picked under a binocular microscope. A total of 100 zircons were mounted in epoxy resin and well-polished to expose grain midsections at about 2/3 to 1/2 of their width. Cathodoluminescent (CL) images were taken to understand grain morphologies and internal structure for in situ analysis (Supplementary Figure 5). U–Pb isotopic analyses for zircons were conducted at the Department of Earth Sciences, The University of Hong Kong. Data were obtained by using a Nu Instruments Multiple Collector (MC) ICP-MS with a Resonetics RESOlution M-50-HR Excimer Laser Ablation System. A beam with energy density of 5 J/cm² hit on sample surface for

ca. 25 s to reach the depth of 20–30 μm. The standard zircons 91500[28] and GJ-1[29] were measured for further calibration. The zircon 91500 was used as an external calibration standard to evaluate the magnitude of mass bias and inter-elemental fractionation. The zircon GJ-1 was used to evaluate the accuracy and precision of the laser-ablation results. The software ICPMSDataCal. Version 8.0[30] was used for calibration and correction. We also used a function given in Anderson[31] for common Pb correction in Microsoft Excel. Tera-Wasserburg and rank order plots were created using ISOPLOT/Excel version 3.0[32]. In this study, 25 zircon grains were randomly selected from the sample, so the results are expected to reflect the characteristics of the age populations. $^{206}Pb/^{238}U$ ages were cited for zircon grains younger than 1000 Ma and $^{207}Pb/^{206}Pb$ ages for older grains. Analyses of not less than 98% concordant were considered for interpreting the age. U–Pb data results are presented in Supplementary Table 1.

**Ammonites**. Three specimens (Supplementary Figure 6) were examined dry under a Nikon SMZ1000 stereomicroscope, and photographs were taken using a Canon 5D digital camera. All specimens (NIGP168515–168517) are deposited in the Nanjing Institute of Geology and Palaeontology, Chinese Academy of Sciences (NIGPAS).

**Pyrolysis gas chromatography mass spectrometry (Py-GC-MS)**. The amber (<0.5 mg of sample) was pyrolyzed at 650 °C for 0.2 min using EGA/PY-3030D Frontier Lab pyrolyser. The pyrolyser was coupled to an Agilent-7890B GC connected with an Agilent-5977B MS. The GC was operated in the splitless mode, equipped with HP-5MS column with dimensions 30 m × 250 μm × 0.25 μm. The GC oven was initially programmed at 40 °C for 1 min and ramped up to 300 °C at 4 °C/min with a final hold of 5.5 min. The initial solvent delay was 2 min. Helium was used as a carrier gas with flow rate of 1 ml/min. The detector source was at 300 °C in EI-mode at 70 eV ionization energy. Spectra were acquired from m/z 50–600. Peak assignments are based on GC retention time and mass spectral data. The analysis was processed at the Department of Earth Sciences, IIT Bombay, Powai, India. The major compounds are presented in Supplementary Table 2.

**Amber inclusions**. We used a Zeiss Stereo Discovery V16 microscope system with Zen software to obtain images. Generally, incident and transmitted light were used simultaneously. To get the clear 3D structures, all images were taken by using digitally stacked photomicrographic composites of ~40 individual focal planes. The plates were made using the image-editing software CorelDraw X7 and Adobe Photoshop CS6. All specimens (NIGP168500–168514) are deposited in the collections of NIGPAS.

**Data availability**. All data generated during this study are included in this published article (and its Supplementary Information files).

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

## Acknowledgements

We are grateful to D. Azar, B.E. Boudinot, M.S. Engel, A.P. Rasnitsyn, F. Stebner, J. Szwedo, and P. Vrsansky for helpful discussions; M. Poujol for advice on the U–Pb dating interpretation; and H. Dong for fieldtrip guidance. This research was supported by the HKU Seed Funding Program for Basic Research (201411159057), National Natural Science Foundation of China (41572010, 41622201, 41688103), and the Strategic Priority Research Program (B) of the Chinese Academy of Sciences (XDB26000000).

## Author contributions

D.Z., S.-C.C. and B.W. designed the project; D.Z., S.-C.C., V.P. and B.W. drafted the manuscript. D.Z., Y.F. and B.W. carried out the geological investigations and collected the specimens. D.Z., V.P., L.M., U.T., S.L., Q.Z., Q.Q.Z., H.W., H.Z. and B.W. identified the amber inclusions and ammonites, and produced the description. D.Z., S.C.C., J. Wong, and J. Wang ran LA-(MC)-ICP-MS analysis and collected the data. S.D. and A.R. ran Py-GC-MS analysis and collected data. V.P., L.M., Q.Z. and Q.Q.Z. produced the fossil photographs, and D.Z. made the plates.

## Additional information

**Competing interests:** The authors declare no competing interests.

