## [Peer Review File · Nature Communications]

Reviewers' comments:

Reviewer #1 (Remarks to the Author):

NCOMMS-18-01983-T

Review of Zheng, Chang et al., A latest Cretaceous amber biota from central Myanmar

The authors report an exceptional assemblage of quite well-preserved species in amber. Their age control above the amber is 71.7 ± 0.25 Ma (youngest age, 76 Ma oldest age) and an ammonite taxon below is known to originate in the Cenomanian (~ 93 Ma) and is common through the Maastrichtian. Their Fig. 1 shows the Tilin amber deposition between Turonian to late Campanian (90-76 Ma). These age estimates show the inherent uncertainties in their data. Illustrations of the amber assemblage are beautiful and well worth publication for their own merits. But the current write up of the data is highly disturbing because of its misrepresentation, numerous errors, erroneous and misleading claims that contradict their data, the constantly claimed but erroneous link to the KPg mass extinction, and question regarding the claimed accuracy of their laser ablation data (see below). All of these are serious issues that misrepresent their findings and interpretations are not supported by their data. This paper comes close to "fake news". For these reasons, I recommend reject. In case the authors rewrite and resubmit for review an article that reflects the actual data and do not make spurious unsupported connections to the end-Cretaceous mass extinction, misrepresent and oversell their data, this paper could be a valuable contribution.

Specific comments and suggestions:

Accuracy of U-Pb age:

The central theme of this paper and all age interpretations of the Tilin amber depend on the accuracy of the LA-ICP-MS zircon geochronology. Authors estimate 71.7 ± 0.25 Ma for the youngest age and older ages of 76 Ma (late Campanian), which they dismissed. The younger age of 71.7 Ma (base Maastrichtian) is probably too precise for laser ablation data, which compared to Tims data often ends up being inaccurate. The reason is that there is no way to account for the Pb-loss, which shifts the age to younger than it actually is. The only way to know whether the ash layer above the Tilin amber is base Maastrichtian or late Campanian is to redo the analyses using Tims.

The current age estimate of 71.7 ± 0.25 Ma is based on a pretty subjective interpretation of the data. It appears that the authors use the histogram in part A of Extended Figure 5 to justify their selection of the nine youngest grains. However, the width of the bins (0.25 Ma) in this histogram is considerably smaller than the reported 2-sigma uncertainties (~ 2 Ma). Consequently, the gap between their yellow histogram bars and teal histogram bars is largely artificial. In part B of this figure they only show the grains used for the age determination which I also find misleading, because the 73 Ma grains would overlap substantially with the reported data set. In other words, the date could easily shift several hundred thousand to a million years older just based on which grains are selected for use in a weighted mean. This would make the sample latest Campanian and not earliest Maastrichtian.

There is also the issue of Pb-loss, which LA-ICP-MS zircon geochronology does not deal with and would result in individual analyses that are younger than the depositional age. In short, I think that the data is probably fine, but that the interpretation is completely subjective. For this MS it is critically important that the site is Maastrichtian and not Campanian in age and for this age determination their laser ablation treatment is not adequate. I'd also like to call attention to their data table that needs to be fixed so that significant digits are handled appropriately and so that the data for reference materials and unknowns are presented in an easy to digest manner.

Errors and misrepresentation of data:

Throughout the paper the authors link the Tilin amber assemblage to the end-Cretaceous mass extinction at 66 Ma for which there is no evidence. They use language such as, the Tilin amber was generated shortly before the KPg mass extinction. But Fig. 1 shows the stratigraphic position of this amber spanning from ~90 to 76 Ma, which is by no means anywhere near the mass extinction. In addition, there are numerous errors in their age estimates, as for example their repeated claim of a 24 m.y. gap between the Maastrichtian and Paleocene. The Maastrichtian through Paleocene age interval spans 16 m.y. and not 24 m.y. Throughout the paper they discuss Cretaceous age intervals by name (e.g., Barremian, Albian, Turonian Coniacian, Santonian, Campanian, Maastrichtian) but never provide the ages for these intervals, although ages are given in Fig. 1. The authors should provide the age for each interval mentioned in the text, which would give the reader the needed evidence to place their faunal claims in the right age context and eliminate erroneous and misleading references to the KPg extinction.

Repetition in writing:

There is significant repetition in the presentation and discussion of faunal occurrences that should be eliminated. The lengthy discussion of the ammonite *Sphenodiscus* found below the Tilin amber is unwarranted given that so little is known, except that this taxon ranges from the Cenomanian through the Maastrichtian, which adds nothing to the age control of the Tilin amber.

Abstract: lines 22-24: Despite the large number of Cretaceous and Cenozoic insect faunas, there is a 24-million-year gap spanning from the Maastrichtian to the Paleocene, which dramatically hinders our understanding of the impact of the K-Pg extinction event on insects.

This sentence makes no sense. The Maastrichtian spans from 72.1 to 66.00 Ma, or 6 m.y. that would leave 18 m.y. missing for the Cenozoic, which brings it well into the middle Eocene. The entire gap would thus span from the Campanian to the middle Eocene.

Lines 25-26: Here, we report a unique amber biota from the Upper Cretaceous (~71.7 Ma) of Tilin, central Myanmar generated shortly before the K-Pg extinction event.

This sentence at best totally misrepresents the results. If the laser ablation age is correct, which is questionable due to unaccounted Pb-loss (see above), the age of 71.7 ± 0.25 Ma is essentially equivalent to the Campanian/Maastrichtian boundary and about 6 Ma before the mass extinction. But given their own laser ablation ages, which exclude older ages of 73-76

Ma, and do not account for Pb-loss (see discussion above), the age of the ash layer above the Tilin amber is most likely late Campanian. This makes the Tilin amber older than late Campanian, which by no definition can be called "shortly before the K-Pg extinction event."

Lines 62-63: Sphenodiscid ammonites were widespread during the Maastrichtian. This is correct. But the authors show this ammonite evolving in the Cenomanian and given that the Tilin amber fauna predates the Maastrichtian and has a depositional age of between Turonian to late Campanian, the discussion of this ammonite in the Maastrichtian has no relevance in this context.

Lines 108 to 113: "Interestingly, some fossils represent the latest known occurrence of typically Cretaceous taxa, demonstrating the survival of these lineages close to the end-Cretaceous: the biting midge (Ceratopogonidae) genus *Protoculicoides* was known from Albian Spanish, Cenomanian French and Kachin, Turonian New Jersey, Coniacian-Santonian Taimyr, and Campanian Canadian ambers; and the mantis (Mantodea) genus *Burmantis* only occurred in Barremian Lebanese and Kachin ambers. These findings further confirm the Cretaceous age of Tilin amber."

The first part of the sentence in the above paragraph is simply false and misleading: the Tilin amber fauna does not demonstrate the survival of these lineages close to the end-Cretaceous given the uncertain age between middle Cenomanian to Campanian (~90-76 Ma, Fig. 1). The second part simply confirms that all the data is from Cretaceous sediments much older than Maastrichtian.

Lines 114-115: "The new Maastrichtian records also comprise lineages of more modern appearance, demonstrating a pre-Cenozoic transition from extinct, stem groups to extant, crown groups."

This sentence is also incorrect and misrepresents the data of this paper. The youngest laser ablation age from above the Tilin amber is 71.7 ± 0.25 Ma, which places it at the Campanian/Maastrichtian boundary (although this age is likely too young due to unaccounted Pb-loss). The Tilin amber is below this age and deposited anytime between Turonian to Campanian (Fig. 1). Therefore, the authors do not have "new Maastrichtian records of modern lineages" but rather show evolution already occurred during the Cenomanian through late Campanian, which is well documented in the literature.

Lines 127-128: "After a 24-million-year gap spanning from Maastrichtian to Paleocene, there is a marked change in the composition and number of ants in fossil deposits."

This sentence is again highly misleading, just plain wrong and fake news. There is no direct age control for the Tilin amber and the depositional age given in Fig. 1 is Turonian to late Campanian, which agrees with the likely laser ablation age (see above). Inferring a Maastrichtian age for the Tilin amber based on the youngest laser ablation age of the overlying ash and ignoring the older late Campanian age for this ash layer is unjustified (see above). Furthermore, the Paleocene spans 10 m.y. from 56-66 Ma. This means that the gap spanning from the base Maastrichtian to Paleocene is 16 m.y. and not 24 m.y. as claimed.

Line 134: "the turnover from stem-groups to crown groups had already begun prior to the K-Pg extinction event."

While this is well documented in the literature, this paper provides no additional support for this claim as the Tilin amber fauna is much older than Maastrichtian age.

Lines 141-143: "Its proximity to the K-Pg extinction event makes the deposit an important point for comparisons across the event, as well as to previous assemblages within the Cretaceous."

This concluding sentence not only overstates the results but is absolutely not supported by the data presented. An amber assemblage with uncertain deposition between Turonian to Campanian cannot be described as "proximity to the K-Pg extinction event" or "an important point of comparison across this (extinction) event".

Reviewer #2 (Remarks to the Author):

Zheng et al. report a very important cretaceous amber outcrop from Myanmar supposedly dated to -71,7 My. These period is crucial to understand the pre-K-Pg event and fill a 24 My gap for insect palaeocenosis.

This gap is very important for our knowledge of the evolution of insect biocenoses linked to the ecological upheavals of the late Cretaceous and the K-Pg crisis.

While it is likely that the K-Pg crisis did not have a very visible impact on the diversity of insects, this important lack of information from this period (no fossil record in insects during 24 My), strongly impeded solid interpretation.

It is now well established from numerous deposits of amber and cretaceous compression fossils that many modern insect lineages were already in place in the middle of the Cretaceous, and few lineages (families or group of families) can be considered extinct. Hence the unsurprising presence of some of them in this new Maastrichtian amber. It is important to stress the need to have an image as close as possible before and after mass extinction events in order to understand its ecological and evolutionary impact. This deposit, well documented by the authors (geology, chemistry of amber) seems to have a very interesting diversity for a first overview and original information on ant lineages and evolution.

The authors present several datings and analytic methods to characterize this new amber, and use also some taxa (Protoculicoides, Burmantis) for confirm the cretaceous age of the inclusions

The presence of ants with 20% of the inclusions (versus 2%) is indeed remarkable but it is necessary to modulate this value which should change with the study of additional materials. Indeed it is difficult to compare amber with each other because of the sampling, the disparity of the deposits (same or different stratigraphic levels ?, taphomomy, botanical origin, etc.) And even more in a preliminary work on a few kilograms of raw amber. In the same way it seems difficult to comment on the absence of taxa, such as sphecomyrmine ants in a small sample.

This will deserve to be verified in future studies, such as those related to the taxonomy of inclusions.

It is very interesting to have precise information on the geology and method of amber extraction, which is not always the case, and an effort has been made in this direction in this manuscript.

Figure 4 poses quality problems (inclusions too dark, visible in the form of silhouettes as if only the diascopic lighting had been used?) Which is not the case of the figure 4, it is a pity to have this difference in the quality of illustration, figure 3 is the more important as it illustrates the taxa commented by the authors and are used to analyze this new palaeocoenosis.

In conclusion, the Maastrichtian amber of Tilin is a major discovery for the paleoecology of the Meso-Cenozoic transition and the evolutionary history of insect and arthropod lineages, and fills a real gap in the fossil record. This discovery, well established in the work of Zheng et al. deserves to be published in Nat. Comm. with some minor revisions proposed (see text comments)

Reviewer #3 (Remarks to the Author):

The authors state that the chemical composition of Tilin amber suggests a tree source among conifers, suggesting that gymnosperms were still abundant in Maastrichtian equatorial forests, and the replacement of gymnosperms by dipterocarps in Southeast Asian tropical forests most likely occurred after the Cretaceous. The results obtained on the chemical characterization of the unique amber biota from the Upper Cretaceous of Tilin look reliable and support the authors statement.

In addition, the composition and number of ants in Tilin amber confirms that tropical forests were the cradle for crown group ants.

Results are of sure interest to other research communities.

My opinion is that the paper can be accepted for publication.

Summary of Comments from Reviewer 2 on Article File

Page: 3

Author: [redacted] Subject: Barrer Date: 2/21/2018 8:06:11 AM -05'00'

Page: 5

Author: [redacted] Subject: Note Date: 2/21/2018 8:21:21 AM -05'00'
you should also consider a dominance of these orders in the forest insect palaeoecenosis

Page: 7

Author: [redacted] Subject: Note Date: 2/22/2018 7:55:13 AM -05'00'
or a different forest in a different ecosystem ? (due to altitude, latitude, etc. effects)

Page: 14

Author: [redacted] Subject: Note Date: 2/22/2018 8:01:46 AM -05'00'
most of the inclusions pictures need to be improved (using Photoshop for reducing contrast and shadows)

Page: 17

Author: [redacted] Subject: Note Date: 2/22/2018 8:14:53 AM -05'00'
just precise here the position of the Kabaw formation and not the Tilin amber
(to be more clear, with brackets like the one of the Upper Cretaceous)

Page: 19

Author: [redacted] Subject: Note Date: 2/22/2018 8:20:57 AM -05'00'
why the inclusions are so dark ?
(lighting problem and post production)
figure 4 don't have this problem

Reviewer 1

NCOMMS-18-01983-T

Review of Zheng, Chang et al., A latest Cretaceous amber biota from central Myanmar

[1] The authors report an exceptional assemblage of quite well-preserved species in amber. Their age control above the amber is 71.7 ± 0.25 Ma (youngest age, 76 Ma oldest age) and an ammonite taxon below is known to originate in the Cenomanian (~93 Ma) and is common through the Maastrichtian. Their Fig. 1 shows the Tilin amber deposition between Turonian to late Campanian (90-76 Ma). These age estimates show the inherent uncertainties in their data. Illustrations of the amber assemblage are beautiful and well worth publication for their own merits. But the current write up of the data is highly disturbing because of its misrepresentation, numerous errors, erroneous and misleading claims that contradict their data, the constantly claimed but erroneous link to the KPg mass extinction, and question regarding the claimed accuracy of their laser ablation data (see below). All of these are serious issues that misrepresent their findings and interpretations are not supported by their data. This paper comes close to “fake news”. For these reasons, I recommend reject. In case the authors rewrite and resubmit for review an article that reflects the actual data and do not make spurious unsupported connections to the end-Cretaceous mass extinction, misrepresent and oversell their data, this paper could be a valuable contribution.

Reply: Thank you very much for your suggestions about our data calculation. We have recalculated our age data, and tried to solve these concerns. However, we don't agree with two vital comments: the age range of the ammonite and the age implications obtained from Fig.1. In our manuscript, we reported that the ammonite *Sphenodiscus* originated from the late Campanian (See lines 76-77) and was widely distributed during Maastrichtian (line 77-78), not originated in the Cenomanian. The previous Fig. 1 probably caused your confusion but it has no such age implications, i.e., Turonian to late Campanian (90-76 Ma). To make it clear, we have marked the Kabaw Formation in the Upper Cretaceous in revised Fig. 1b. The modification clearly shows that the age of Tilin amber is latest Campanian.

We have never directly linked this amber biota to the KPg mass extinction. We demonstrate that Tilin amber biota yields the latest known diverse insect assemblage in the Mesozoic, and partly fills the 24-million-year palaeoentomological gap. We are curious which parts of this manuscript mention the link between amber biota and mass extinction. Please provide line numbers or page numbers so we can revise it.

For the other suggestions, please find detailed responses below.

Specific comments and suggestions:

Accuracy of U-Pb age:

[2] The central theme of this paper and all age interpretations of the Tilin amber depend on the accuracy of the LA-ICP-MS zircon geochronology. Authors estimate 71.7 ± 0.25 Ma for the youngest age and older ages of 76 Ma (late Campanian), which they dismissed. The younger age of 71.7 Ma (base Maastrichtian) is probably too precise for laser ablation data, which compared to Tims data often ends up being inaccurate. The reason is that there is no way to account for the Pb-loss, which shifts the age to younger than it actually is. The only way to know whether the ash layer above the Tilin amber is base Maastrichtian or late Campanian is to redo the analyses using Tims.

Reply: Please note that our equipment is LA-(MC)-ICP-MS (see Line 173), instead of LA-ICP-MS. Much studies have proved that LA-(MC)-ICP-MS zircon dating gives precise and accurate ages (e.g. Gerdes & Zeh, 2006; Simonetti et al., 2005, 2006; Gehrels et al., 2008; Cottle et al., 2009; Fassett et al., 2011). Very recently, Lana et al. (2017) indicated the enhanced sensitivity and stability of the LA-(MC)-ICP-MS system has allowed the determination of apparent Pb/U and Pb/Pb ratios with uncertainties of 0.3 to 1% (2 sigma errors). We agree that ID-TIMS will provide an accurate age determination. However, we believe that our age result obtained by LA-(MC)-ICP-MS U/Pb dating is reliable and precise. In addition to radio-isotopic data, the age of this biota is also constrained by the Late Campanian-Maastrichtian ammonites *Sphenodiscus*. Our absolute age data is in a good agreement with the relative age.

On the other hand, we have tried to avoid the age data with Pb-loss in the revised manuscript. We have made a Tera-Wasserburg diagram, which shows no or weak Pb-loss. Pb-loss is certainly not an issue in this study. Nearly all of our analyses have ages around 72 Ma. We used ten data from the youngest age population to calculate the weighed mean age, representing the maximum depositional age of our specimen. This revised age is convincing and reliable.

[3] The current age estimate of 71.7 ± 0.25 Ma is based on a pretty subjective interpretation of the data. It appears that the authors use the histogram in part A of Extended Figure 5 to justify their selection of the nine youngest grains. However, the width of the bins (0.25 Ma) in this histogram is considerably smaller than the reported 2-sigma uncertainties (~2 Ma). Consequently, the gap between their yellow histogram bars and teal histogram bars is largely artificial. In part B of this figure they only show the grains used for the age determination which I also find misleading, because the 73 Ma grains would overlap substantially with the reported data set. In other words, the date could easily shift several hundred thousand to a million years older just based on which grains are selected for use in a weighted mean. This would make the sample latest Campanian and not earliest Maastrichtian.

Reply: Thanks for this suggestion. We have recalculated the data. We agree that the width of the bins should be changed to 2 Ma, to be consistent with the uncertainties. However, we deleted this diagram since that Tera-Wasserburg diagram and histograms are good enough to reflect our dating result. In the revised manuscript, we put all data on the Tera-Wasserburg diagram

(Supplementary Fig. 5a) to identify any potential Pb-loss. We interpret 72.2 ± 0.3 Ma (latest Campanian age) as the maximum depositional age for the Tilin amber.

[4] There is also the issue of Pb-loss, which LA-ICP-MS zircon geochronology does not deal with and would result in individual analyses that are younger than the depositional age. In short, I think that the data is probably fine, but that the interpretation is completely subjective. For this MS it is critically important that the site is Maastrichtian and not Campanian in age and for this age determination their laser ablation treatment is not adequate. I'd also like to call attention to their data table that needs to be fixed so that significant digits are handled appropriately and so that the data for reference materials and unknowns are presented in an easy to digest manner.

Reply: Thank you very much for your suggestions. For the Pb-loss concern, please see reply for Comment [2]. The site is considered as the latest Campanian in the revised manuscript. As suggested by the reviewer, we fixed the data table.

[5] Errors and misrepresentation of data:

Throughout the paper the authors link the Tilin amber assemblage to the end-Cretaceous mass extinction at 66 Ma for which there is no evidence. They use language such as, the Tilin amber was generated shortly before the KPg mass extinction. But Fig. 1 shows the stratigraphic position of this amber spanning from ~90 to 76 Ma, which is by no means anywhere near the mass extinction. In addition, there are numerous errors in their age estimates, as for example their repeated claim of a 24 m.y. gap between the Maastrichtian and Paleocene. The Maastrichtian through Paleocene age interval spans 16 m.y. and not 24 m.y. Throughout the paper they discuss Cretaceous age intervals by name (e.g., Barremian, Albian, Turonian Coniacian, Santonian, Campanian, Maastrichtian) but never provide the ages for these intervals, although ages are given in Fig. 1. The authors should provide the age for each interval mentioned in the text, which would give the reader the needed evidence to place their faunal claims in the right age context and eliminate erroneous and misleading references to the KPg extinction.

Reply: About Fig.1 and its age implications, please see Comment [2]. The 24 Myr gap has been indicated in the Supplementary Information (see Lines 28-31 in SM) and should span from the early Campanian to the early Eocene. We did not add the age intervals, since they have been given in revised Fig. 1.

Repetition in writing:

[6] There is significant repetition in the presentation and discussion of faunal occurrences that should be eliminated. The lengthy discussion of the ammonite *Sphenodiscus* found below the Tilin amber is unwarranted given that so little is known, except that this taxon ranges from the Cenomanian through the Maastrichtian, which adds nothing to the age control of the Tilin amber.

Reply: The range of *Sphenodiscus* was stated from late Campanian (please see Comment [2]).

The discussion about the ammonite is necessary because it provides the key evidence about its biostratigraphic age.

[7] Abstract: lines 22-24: Despite the large number of Cretaceous and Cenozoic insect faunas, there is a 24-million-year gap spanning from the Maastrichtian to the Paleocene, which dramatically hinders our understanding of the impact of the K-Pg extinction event on insects. This sentence makes no sense. The Maastrichtian spans from 72.1 to 66.00 Ma, or 6 m.y. that would leave 18 m.y. missing for the Cenozoic, which brings it well into the middle Eocene. The entire gap would thus span from the Campanian to the middle Eocene.

Reply: Thanks. We have rephrased that this gap should span from the early Campanian to the early Eocene. Please see lines 40-41, 143.

[8] Lines 25-26: Here, we report a unique amber biota from the Upper Cretaceous (~71.7 Ma) of Tilin, central Myanmar generated shortly before the K-Pg extinction event. This sentence at best totally misrepresents the results. If the laser ablation age is correct, which is questionable due to unaccounted Pb-loss (see above), the age of 71.7 ± 0.25 Ma is essentially equivalent to the Campanian/Maastrichtian boundary and about 6 Ma before the mass extinction. But given their own laser ablation ages, which exclude older ages of 73-76 Ma, and do not account for Pb-loss (see discussion above), the age of the ash layer above the Tilin amber is most likely late Campanian. This makes the Tilin amber older than late Campanian, which by no definition can be called “shortly before the K-Pg extinction event.”

Reply: Thanks. We delete ‘shortly’ to make it more accurate. Please see Line 23.

[9] Lines 62-63: Sphenodiscid ammonites were widespread during the Maastrichtian. This is correct. But the authors show this ammonite evolving in the Cenomanian and given that the Tilin amber fauna predates the Maastrichtian and has a depositional age of between Turonian to late Campanian, the discussion of this ammonite in the Maastrichtian has no relevance in this context.

Reply: About the evolving of this ammonite, please see the response in Comment [2].

[10] Lines 108 to 113: “Interestingly, some fossils represent the latest known occurrence of typically Cretaceous taxa, demonstrating the survival of these lineages close to the end-Cretaceous: the biting midge (Ceratopogonidae) genus *Protoculicoides* was known from Albian Spanish, Cenomanian French and Kachin, Turonian New Jersey, Coniacian-Santonian Taimyr, and Campanian Canadian ambers; and the mantis (Mantodea) genus *Burmantis* only occurred in Barremian Lebanese and Kachin ambers. These findings further confirm the Cretaceous age of Tilin amber.”

The first part of the sentence in the above paragraph is simply false and misleading: the Tilin amber fauna does not demonstrate the survival of these lineages close to the end-Cretaceous given the uncertain age between middle Cenomanian to Campanian (~90-76 Ma, Fig. 1). The

second part simply confirms that all the data is from Cretaceous sediments much older than Maastrichtian.

Reply: We don't agree with this comment and kept our claim. The age of amber biota was well given and the Fig.1 has no such age implications (please see Comment [2]).

[11] Lines 114-115: “The new Maastrichtian records also comprise lineages of more modern appearance, demonstrating a pre-Cenozoic transition from extinct, stem groups to extant, crown groups.”

This sentence is also incorrect and misrepresents the data of this paper. The youngest laser ablation age from above the Tilin amber is 71.7 ± 0.25 Ma, which places it at the Campanian/Maastrichtian boundary (although this age is likely too young due to unaccounted Pb-loss). **The Tilin amber is below this age and deposited anytime between Turonian to Campanian (Fig. 1).** Therefore, the authors do not have “new Maastrichtian records of modern lineages” but rather show evolution already occurred during the Cenomanian through late Campanian, which is well documented in the literature.

Reply: Please see our response to Comment [2].

[12] Lines 127-128: “After a 24-million-year gap spanning from Maastrichtian to Paleocene, there is a marked change in the composition and number of ants in fossil deposits.”

This sentence is again highly misleading, just plain wrong and fake news. **There is no direct age control for the Tilin amber and the depositional age given in Fig. 1 is Turonian to late Campanian, which agrees with the likely laser ablation age (see above).** Inferring a Maastrichtian age for the Tilin amber based on the youngest laser ablation age of the overlying ash and ignoring the older late Campanian age for this ash layer is unjustified (see above). Furthermore, the Paleocene spans 10 m.y. from 56-66 Ma. This means that the gap spanning from the base Maastrichtian to Paleocene is 16 m.y. and not 24 m.y. as claimed.

Reply: Thanks. We has changed ‘Maastrichtian to Paleocene’ to ‘the early Campanian to the early Eocene’ (the evidence of 24 Myr has provided in Comment [5]). Please see Line 143.

[13] Line 134: “the turnover from stem-groups to crown groups had already begun prior to the K-Pg extinction event.”

While this is well documented in the literature, this paper provides no additional support for this claim as the Tilin amber fauna is much older than Maastrichtian age.

Reply: Thanks for the comments. However, in the revised manuscript, a latest Campanian age for this amber biota still supports this claim.

[14] Lines 141-143: “Its proximity to the K-Pg extinction event makes the deposit an important point for comparisons across the event, as well as to previous assemblages within the Cretaceous.”

This concluding sentence not only overstates the results but is absolutely not supported by the data presented. An amber assemblage with uncertain deposition between Turonian to Campanian

cannot be described as “proximity to the K-Pg extinction event” or “an important point of comparison across this (extinction) event”.

Reply: We have given this amber biota a reliable latest Campanian age after robust isotopic age (72.3 Ma) and ammonites (originated from Late Campanian and widely distributed during Maastrichtian). The latest Campanian age is proximity to the K-Pg extinction event, and has these implications.

Reviewer 2

Zheng et al. report a very important cretaceous amber outcrop from Myanmar supposedly dated to -71,7 My. These period is crucial to understand the pre-K-Pg event and fill a 24 My gap for insect palaeocenosis.

This gap is very important for our knowledge of the evolution of insect biocenoses linked to the ecological upheavals of the late Cretaceous and the K-Pg crisis.

While it is likely that the K-Pg crisis did not have a very visible impact on the diversity of insects, this important lack of information from this period (no fossil record in insects during 24 My), strongly impeded solid interpretation.

It is now well established from numerous deposits of amber and cretaceous compression fossils that many modern insect lineages were already in place in the middle of the Cretaceous, and few lineages (families or group of families) can be considered extinct. Hence the unsurprising presence of some of them in this new Maastrichtian amber. It is important to stress the need to have an image as close as possible before and after mass extinction events in order to understand its ecological and evolutionary impact. This deposit, well documented by the authors (geology, chemistry of amber) seems to have a very interesting diversity for a first overview and original information on ant lineages and evolution.

The authors present several datings and analytic methods to characterize this new amber, and use also some taxa (Protoculicoides, Burmantis) for confirm the cretaceous age of the inclusions.

Reply: We greatly appreciate the reviewer’s comments regarding the remarkable nature of the current find. We totally agree with the reviewer’s suggestions.

[15] The presence of ants with 20% of the inclusions (versus 2%) is indeed remarkable but it is necessary to modulate this value which should change with the study of additional materials.

Reply: Thanks. We have added ‘but more Tilin amber samples are required for a more accurate estimate’. Please see lines 147-148.

Indeed it is difficult to compare amber with each other because of the sampling, the disparity of the deposits (same or different stratigraphic levels? taphonomy, botanical origin, etc.) And even more in a preliminary work on a few kilograms of raw amber. In the same way it seems difficult to comment on the absence of taxa, such as sphecomyrmine ants in a small sample.

This will deserve to be verified in future studies, such as those related to the taxonomy of

inclusions.

It is very interesting to have precise information on the geology and method of amber extraction, which is not always the case, and an effort has been made in this direction in this manuscript.

[16] Figure 4 poses quality problems (inclusions too dark, visible in the form of silhouettes as if only the diascopic lighting had been used?) Which is not the case of the figure 4, it is a pity to have this difference in the quality of illustration, figure 3 is the more important as it illustrates the taxa commented by the authors and are used to analyze this new palaeocoenosis.

Reply: Thanks. We have polished these specimens again and taken new photos for most specimens. These photos were also readjusted by adding lighting and reducing contrasting. Please see the revised Figure 3.

In conclusion, the Maastrichtian amber of Tilin is a major discovery for the paleoecology of the Meso-Cenozoic transition and the evolutionary history of insect and arthropod lineages, and fills a real gap in the fossil record. This discovery, well established in the work of Zheng et al. deserves to be published in Nat. Comm. with some minor revisions proposed (see text comments).

[17] Line 58 Delete 'specific'.

Reply: Thanks. Done. Please see Line 70.

[18] Line 102 You should also consider a dominance of these orders in the forest insect palaeocoenosis.

Reply: Thanks. Done. Please see Line 117.

[19] Line 145 or a different forest in a different ecosystem? (Due to altitude, latitude, etc. effects)

Reply: Thanks. We have added the role of tropical forest. Please see Line 117.

[20] Line 314 most of the inclusions pictures need to be improved (using Photoshop for reducing contrast and shadows)

Reply: Thanks. Done. Please see comment [18].

[21] Line 359. Figure 1: just precise here the position of the Kabaw formation and not the Tilin amber (to be clearer, with brackets like the one of the Upper Cretaceous).

Reply: Thanks. We have put the Kabaw Formation in a bracket, showing its position in the Upper Cretaceous. Please see revised Fig. 1.

[22] Line 390. Figure 3: why the inclusions are so dark? (Lighting problem and post production). Figure 4 don't have this problem.

Reply: Thanks. Done. Please see comment [18].

Reviewer 3

The authors state that the chemical composition of Tilin amber suggests a tree source among conifers, suggesting that gymnosperms were still abundant in Maastrichtian equatorial forests, and the replacement of gymnosperms by dipterocarps in Southeast Asian tropical forests most likely occurred after the Cretaceous.

The results obtained on the chemical characterization of the unique amber biota from the Upper Cretaceous of Tilin look reliable and support the authors statement.

In addition, the composition and number of ants in Tilin amber confirms that tropical forests were the cradle for crown group ants.

Results are of sure interest to other research communities.

My opinion is that the paper can be accepted for publication.

Reply: Thank you very much for your positive comments of this manuscript.

Reviewer 1

NCOMMS-18-01983-T

Review of Zheng, Chang et al., A latest Cretaceous amber biota from central Myanmar

Review of revised version manuscript NCOMMS-18-01983A

The authors have made substantial progress in their revision to eliminate misrepresentations and unsupported claims and statements particularly with respect to the Tilin fauna being close to the KPg or telling us how this fauna gives insight into what happened at the KPg. But the revision is sloppy and some parts of the MS still keep the misrepresentation and missinterpretations claiming a connection to the late Campanian Tilin fauna. This is simply nonsense because the mass extinction is more than 6 m.y. later and follows Deccan volcanism that changed the world's climate and environment ending with the mass extinction. A Campanian age therefore cannot be considered giving insight into the KPg mass extinction. The authors have reconisidered their UPb age and changed it accordingly to late Campanian, which we suggested is the best age given their data and metods. But they provide no methods or rational for chosing certain ages and not others to reach this conclusion, nor have they adjusted the figures in the supplementary data to reflect this change, or written detailed methods. I recommend rejection of this MS, but am confident that the authors can easily made the necessary revisions to make this a paper consistent with the data and without misrepresentations of the data and unsupported claims.

Further comments pertaining to this review of the revised MS are given below in blue.

[1] The authors report an exceptional assemblage of quite well-preserved species in amber. Their age control above the amber is 71.7 ± 0.25 Ma (youngest age, 76 Ma oldest age) and an ammonite taxon below is known to originate in the Cenomanian (~93 Ma) and is common through the Maastrichtian. Their Fig. 1 shows the Tilin amber deposition between Turonian to late Campanian (90-76 Ma). These age estimates show the inherent uncertainties in their data. Illustrations of the amber assemblage are beautiful and well worth publication for their own merits. But the current write up of the data is highly disturbing because of its misrepresentation, numerous errors, erroneous and misleading claims that contradict their data, the constantly claimed but erroneous link to the KPg mass extinction, and question regarding the claimed accuracy of their laser ablation data (see below). All of these are serious issues that misrepresent their findings and interpretations are not supported by their data. This paper comes close to “fake news”. For these reasons, I recommend reject. In case the authors rewrite and resubmit for review an article that reflects the actual data and do not make spurious unsupported connections to the end-Cretaceous mass extinction, misrepresent and oversell their data, this paper could be a valuable contribution.

Reply: Thank you very much for your suggestions about our data calculation. We have re-calculated our age data, and tried to solve these concerns. However, we don't agree with two vital comments: the age range of the ammonite and the age implications obtained from Fig.1. In

our manuscript, we reported that the ammonite *Sphenodiscus* originated from the late Campanian (See lines 76-77) and was widely distributed during Maastrichtian (line 77-78), not originated in the Cenomanian. The previous Fig. 1 probably caused your confusion but it has no such age implications, i.e., Turonian to late Campanian (90-76 Ma). To make it clear, we have marked the Kabaw Formation in the Upper Cretaceous in revised Fig. 1b. The modification clearly shows that the age of Tilin amber is latest Campanian.

OK This part of the MS is now clear in the figure and text.

We have never directly linked this amber biota to the KPg mass extinction. We demonstrate that Tilin amber biota yields the latest known diverse insect assemblage in the Mesozoic, and partly fills the 24-million-year palaeontological gap. We are curious which parts of this manuscript mention the link between amber biota and mass extinction. Please provide line numbers or page numbers so we can revise it.

In fact, the main message in the original MS was that the Tilin fauna provides insight into the KPg mass extinction. Yes, never made a direct link because there is none, but you certainly claimed there was and in parts of the revised MS you still push this idea. This misrepresentation must be eliminated in the entire MS and Supplementary data (where it is still there too).

For the other suggestions, please find detailed responses below.

Specific comments and suggestions:

Accuracy of U-Pb age:

[2] The central theme of this paper and all age interpretations of the Tilin amber depend on the accuracy of the LA-ICP-MS zircon geochronology. Authors estimate 71.7 ± 0.25 Ma for the youngest age and older ages of 76 Ma (late Campanian), which they dismissed. The younger age of 71.7 Ma (base Maastrichtian) is probably too precise for laser ablation data, which compared to Tims data often ends up being inaccurate. The reason is that there is no way to account for the Pb-loss, which shifts the age to younger than it actually is. The only way to know whether the ash layer above the Tilin amber is base Maastrichtian or late Campanian is to redo the analyses using Tims.

Reply: Please note that our equipment is LA-(MC)-ICP-MS (see Line 173), instead of LA-ICP-MS. Much studies have proved that LA-(MC)-ICP-MS zircon dating gives precise and accurate ages (e.g. Gerdes & Zeh, 2006; Simonetti et al., 2005, 2006; Gehrels et al., 2008; Cottle et al., 2009; Fassett et al., 2011). Very recently, Lana et al. (2017) indicated the enhanced sensitivity and stability of the LA-(MC)-ICP-MS system has allowed the determination of apparent Pb/U and Pb/Pb ratios with uncertainties of 0.3 to 1% (2 sigma errors). We agree that ID-TIMS will provide an accurate age determination. However, we believe that our age result obtained by LA-(MC)-ICP-MS U/Pb dating is reliable and precise. In addition to radio-isotopic data, the age of this biota is also constrained by the Late Campanian-Maastrichtian ammonites *Sphenodiscus*. Our absolute age data is in a good agreement with the relative age.

On the other hand, we have tried to avoid the age data with Pb-loss in the revised manuscript. We have made a Tera-Wasserburg diagram, which shows no or weak Pb-loss. Pb-loss is certainly not an issue in this study. Nearly all of our analyses have ages around 72 Ma. We used ten data from the youngest age population to calculate the weighed mean age, representing the maximum depositional age of our specimen. This revised age is convincing and reliable.

Reply:

Basically, the data is treated better, but the methods aren't described and the presentation remains sloppy in the supplement. These issues need to be corrected before final acceptance. However, I am confident that you will be able to address them.

Specific problems that still need to be addressed:

The geochronology data is treated somewhat better. I would have included one more older grain in the date interpretation. This interpretation would be more statistically valid, MSWD = 0.6 rather than MSWD of 0.3. However, doing this does not change the date significantly. So, it does not really matter.

The comment that this sample does not contain Pb-loss is difficult to evaluate. Subtle Pb-loss may very well exist. Regardless, Pb-loss will not show up very well on either a Wetherill or a Tera-Wasserburg concordia for rocks of this age. So, the inclusion of this plot does not preclude Pb-loss. Their best argument for this interpretation is that the dates cluster very well.

A more significant problem is that the U-Pb methods are not described in the supplementary material or in the main text (i.e., what machine was used, how were the ratios corrected for instrumental fractionation, what reference materials were used to calibrate corrections, and which reference materials were used to assess external reproducibility?). This information absolutely needs to be in the supplementary material.

Neither reference material used in this study is appropriate for evaluating reproducibility of $^{206}\text{Pb}/^{238}\text{U}$ dates, as they are both (GJ-1 and 91500) intended to be used as reference materials for $^{207}\text{Pb}/^{206}\text{Pb}$ ratios. The $^{207}\text{Pb}/^{206}\text{Pb}$ dates for zircon reference material 91500 report negative uncertainties, which is not meaningful and must reflect a mistake. This issue needs to be corrected.

The caption on the geochronology figure in the supplement still mentions the histogram rather than the tera-wasserburg plot and rank order plot. Also, the uncertainties need to be identified in this caption (1 sigma vs. 2 sigma).

In sum, the data interpretation is better and acceptable and is used to justify a late Campanian age. This age interpretation is more reasonable than the previous interpretation. However, the documentation of the U-Pb geochronology methods and

presentation of the data in the supplementary material needs to be improved before publication. In the long term, I recommend that the laboratory doing these analyses procure a zircon reference material appropriate for calibrating $^{206}\text{Pb}/^{238}\text{U}$ ratios or at least one that is appropriate for testing the external reproducibility of this measurement.

[3] The current age estimate of 71.7 ± 0.25 Ma is based on a pretty subjective interpretation of the data. It appears that the authors use the histogram in part A of Extended Figure 5 to justify their selection of the nine youngest grains. However, the width of the bins (0.25 Ma) in this histogram is considerably smaller than the reported 2-sigma uncertainties (~ 2 Ma). Consequently, the gap between their yellow histogram bars and teal histogram bars is largely artificial. In part B of this figure they only show the grains used for the age determination which I also find misleading, because the 73 Ma grains would overlap substantially with the reported data set. In other words, the date could easily shift several hundred thousand to a million years older just based on which grains are selected for use in a weighted mean. This would make the sample latest Campanian and not earliest Maastrichtian.

Reply: Thanks for this suggestion. We have recalculated the data. We agree that the width of the bins should be changed to 2 Ma, to be consistent with the uncertainties. However, we deleted this diagram since that Tera-Wasserburg diagram and histograms are good enough to reflect our dating result. In the revised manuscript, we put all data on the Tera-Wasserburg diagram (Supplementary Fig. 5a) to identify any potential Pb-loss. We interpret 72.2 ± 0.3 Ma (latest Campanian age) as the maximum depositional age for the Tilin amber.

See detailed comments above

[4] There is also the issue of Pb-loss, which LA-ICP-MS zircon geochronology does not deal with and would result in individual analyses that are younger than the depositional age. In short, I think that the data is probably fine, but that the interpretation is completely subjective. For this MS it is critically important that the site is Maastrichtian and not Campanian in age and for this age determination their laser ablation treatment is not adequate. I'd also like to call attention to their data table that needs to be fixed so that significant digits are handled appropriately and so that the data for reference materials and unknowns are presented in an easy to digest manner.

Reply: Thank you very much for your suggestions. For the Pb-loss concern, please see reply for Comment [2]. The site is considered as the latest Campanian in the revised manuscript. As suggested by the reviewer, we fixed the data table.

See detailed comments above

[5] Errors and misrepresentation of data:

Throughout the paper the authors link the Tilin amber assemblage to the end-Cretaceous mass extinction at 66 Ma for which there is no evidence. They use language such as, the Tilin amber

was generated shortly before the KPg mass extinction. But Fig. 1 shows the stratigraphic position of this amber spanning from ~90 to 76 Ma, which is by no means anywhere near the mass extinction. In addition, there are numerous errors in their age estimates, as for example their repeated claim of a 24 m.y. gap between the Maastrichtian and Paleocene. The Maastrichtian through Paleocene age interval spans 16 m.y. and not 24 m.y. Throughout the paper they discuss Cretaceous age intervals by name (e.g., Barremian, Albian, Turonian Coniacian, Santonian, Campanian, Maastrichtian) but never provide the ages for these intervals, although ages are given in Fig. 1. The authors should provide the age for each interval mentioned in the text, which would give the reader the needed evidence to place their faunal claims in the right age context and eliminate erroneous and misleading references to the KPg extinction.

Reply: About Fig.1 and its age implications, please see Comment [2]. The 24 Myr gap has been indicated in the Supplementary Information (see Lines 28-31 in SM) and should span from the early Campanian to the early Eocene. We did not add the age intervals, since they have been given in revised Fig. 1.

OK for this problem

Repetition in writing:

[6] There is significant repetition in the presentation and discussion of faunal occurrences that should be eliminated. The lengthy discussion of the ammonite *Sphenodiscus* found below the Tilin amber is unwarranted given that so little is known, except that this taxon ranges from the Cenomanian through the Maastrichtian, which adds nothing to the age control of the Tilin amber.

Reply: The range of *Sphenodiscus* was stated from late Campanian (please see Comment [2]). The discussion about the ammonite is necessary because it provides the key evidence about its biostratigraphic age.

I strongly disagree. The long discussion and description of the ammonites is **not necessary**. The only age control the ammonites provide is that they range from the late Campanian to the end Maastrichtian. In this they confirm that the Tilin sample is of Late Campanian age, just as the revised UPb age indicates. Ammonites below the Tilin fauna give no information about ammonites above this fauna or at the KPg. Furthermore, the lengthy ammonite discussion (without any illustrations) is a distraction and digression from the central part of this MS, the Tilin fauna. I strongly recommend that this discussion be relegated to supplementary material where the illustration is, and only a short statement is made in the text that first appearance of ammonites below the Tilin fauna reveal a late Campanian age in accordance with the UPb age derived from above the Tilin fauna.

[7] Abstract: lines 22-24: Despite the large number of Cretaceous and Cenozoic insect faunas, there is a 24-million-year gap spanning from the Maastrichtian to the Paleocene, which dramatically hinders our understanding of the impact of the K-Pg extinction event on insects. This sentence makes no sense. The Maastrichtian spans from 72.1 to 66.00 Ma, or 6 m.y. that

would leave 18 m.y. missing for the Cenozoic, which brings it well into the middle Eocene. The entire gap would thus span from the Campanian to the middle Eocene.

Reply: Thanks. We have rephrased that this gap should span from the early Campanian to the early Eocene. Please see lines 40-41, 143.

OK with this correction, but why was it misrepresented in the first place?

[8] Lines 25-26: Here, we report a unique amber biota from the Upper Cretaceous (~71.7 Ma) of Tilin, central Myanmar generated shortly before the K-Pg extinction event.

This sentence at best totally misrepresents the results. If the laser ablation age is correct, which is questionable due to unaccounted Pb-loss (see above), the age of 71.7 ± 0.25 Ma is essentially equivalent to the Campanian/Maastrichtian boundary and about 6 Ma before the mass extinction. But given their own laser ablation ages, which exclude older ages of 73-76 Ma, and do not account for Pb-loss (see discussion above), the age of the ash layer above the Tilin amber is most likely late Campanian. This makes the Tilin amber older than late Campanian, which by no definition can be called “shortly before the K-Pg extinction event.”

Reply: Thanks. We delete ‘shortly’ to make it more accurate. Please see Line 23.

Deleting “shortly” from “before the KPg extinction event” is insufficient. The Tilin fauna is of late Campanian age and hence it was generated “shortly before the end-Campanian”, which precedes the KPg by over 6 m.y. Any KPg mention must be eliminate as this is gross misrepresentation of your data.

[9] Lines 62-63: Sphenodiscid ammonites were widespread during the Maastrichtian.

This is correct. But the authors show this ammonite evolving in the Cenomanian and given that the Tilin amber fauna predates the Maastrichtian and has a depositional age of between Turonian to late Campanian, the discussion of this ammonite in the Maastrichtian has no relevance in this context.

Reply: About the evolving of this ammonite, please see the response in Comment [2].

See my response above in (6)

[10] Lines 108 to 113: “Interestingly, some fossils represent the latest known occurrence of typically Cretaceous taxa, demonstrating the survival of these lineages close to the end-Cretaceous: the biting midge (Ceratopogonidae) genus *Protoculicoides* was known from Albian Spanish, Cenomanian French and Kachin, Turonian New Jersey, Coniacian-Santonian Taimyr, and Campanian Canadian ambers; and the mantis (Mantodea) genus *Burmantis* only occurred in Barremian Lebanese and Kachin ambers. These findings further confirm the Cretaceous age of Tilin amber.”

The first part of the sentence in the above paragraph is simply false and misleading: the Tilin amber fauna does not demonstrate the survival of these lineages close to the end-Cretaceous given the uncertain age between middle Cenomanian to Campanian (~90-76 Ma, Fig. 1). The

second part simply confirms that all the data is from Cretaceous sediments much older than Maastrichtian.

Reply: We don't agree with this comment and kept our claim. The age of amber biota was well given and the Fig.1 has no such age implications (please see Comment [2]).

Strongly disagree: In the first sentence (now in revised MS 123-127), the youngest age quoted for the fauna is late Campanian, which does not support your statement “demonstrating the survival of these lineages close to the end-Cretaceous”. The latter statement implies it's latest Maastrichtian, when in fact it's late Campanian. This type of misrepresentation has been the problem throughout this MS in the first round and still plagues the revised version. This must be corrected before acceptance. Science writing must be precise and not misrepresent ages with unsupported loose interpretations.

[11] Lines 114-115: “The new Maastrichtian records also comprise lineages of more modern appearance, demonstrating a pre-Cenozoic transition from extinct, stem groups to extant, crown groups.”

This sentence is also incorrect and misrepresents the data of this paper. The youngest laser ablation age from above the Tilin amber is 71.7 ± 0.25 Ma, which places it at the Campanian/Maastrichtian boundary (although this age is likely too young due to unaccounted Pb-loss). The Tilin amber is below this age and deposited anytime between Turonian to Campanian (Fig. 1). Therefore, the authors do not have “new Maastrichtian records of modern lineages” but rather show evolution already occurred during the Cenomanian through late Campanian, which is well documented in the literature.

Reply: Please see our response to Comment [2].

OK. Authors corrected “Maastrichtian” with “Late Campanian” and acknowledged that the Tilin fauna is late Campanian in age.

[12] Lines 127-128: “After a 24-million-year gap spanning from Maastrichtian to Paleocene, there is a marked change in the composition and number of ants in fossil deposits.”

This sentence is again highly misleading, just plain wrong and fake news. There is no direct age control for the Tilin amber and the depositional age given in Fig. 1 is Turonian to late Campanian, which agrees with the likely laser ablation age (see above). Inferring a Maastrichtian age for the Tilin amber based on the youngest laser ablation age of the overlying ash and ignoring the older late Campanian age for this ash layer is unjustified (see above). Furthermore, the Paleocene spans 10 m.y. from 56-66 Ma. This means that the gap spanning from the base Maastrichtian to Paleocene is 16 m.y. and not 24 m.y. as claimed.

Reply: Thanks. We has changed ‘Maastrichtian to Paleocene’ to ‘the early Campanian to the early Eocene’ (the evidence of 24 Myr has provided in Comment [5]). Please see Line 143.

Ok fine

[13] Line 134: “the turnover from stem-groups to crown groups had already begun prior to the K-Pg extinction event.”

While this is well documented in the literature, this paper provides no additional support for this claim as the Tilin amber fauna is much older than Maastrichtian age.

Reply: Thanks for the comments. However, in the revised manuscript, a latest Campanian age for this amber biota still supports this claim.

Quote now in line 150 revised MS:

Strongly disagree. The Tilin fauna is late Campanian in age and predates the KPg by more than 6 m.y. see my suggested revision in annotated MS.

[14] Lines 141-143: “Its proximity to the K-Pg extinction event makes the deposit an important point for comparisons across the event, as well as to previous assemblages within the Cretaceous.” This concluding sentence not only overstates the results but is absolutely not supported by the data presented. An amber assemblage with uncertain deposition between Turonian to Campanian cannot be described as “proximity to the K-Pg extinction event” or “an important point of comparison across this (extinction) event”.

Reply: We have given this amber biota a reliable latest Campanian age after robust isotopic age (72.3 Ma) and ammonites (originated from Late Campanian and widely distributed during Maastrichtian). The latest Campanian age is proximity to the K-Pg extinction event, and has these implications.

Strongly disagree: Latest Campanian age is more than 6 m.y. after the the Campanian and therefore decidedly **NOT proximity to KPg extinction** and has no implications for the KPg mass extinction. This is sloppy writing and continues the same misrepresentation that plagues this study in both the original and revised submissions. **In geology, six m.y. before KPg is definitely not proximal to the mass extinction under any definition. This must be deleted.** The late Campanian Tilin fauna is a good comparison point for older Cretaceous faunas and Cenozoic faunas, but says nothing about what happened at the KPg.

Reviewer 1

Review of revised version manuscript NCOMMS-18-01983A

[1] The authors have made substantial progress in their revision to eliminate misrepresentations and unsupported claims and statements particularly with respect to the Tulin fauna being close to the KPg or telling us how this fauna gives insight into what happened at the KPg. But the revision is sloppy and some parts of the MS still keep the misrepresentation and missinterpretations claiming a connection to the late Campanian Tulin fauna. This is simply nonsense because the mass extinction is more than 6 m.y. later and follows Deccan volcanism that changed the world's climate and environment ending with the mass extinction. A Campanian age therefore cannot be considered giving insight into the KPg mass extinction.

Reply: We truly appreciate your time and efforts on this manuscript. We fully agree with your comments, and we have deleted/revise all unsuitable claims related to the KPg event (or latest Cretaceous). Please see lines 1, 23, 31–33, 39–40, 42, 124, 149, 157, and 159 in the main text, and lines 34–35, 40, 70 and Supplementary Fig. 1a in the Supplementary Information.

[2] The authors have reconisidered their UPb age and changed it accordingly to late Campanian, which we suggested is the best age given their data and methods. But they provide no methods or rationale for choosing certain ages and not others to reach this conclusion, nor have they adjusted the figures in the supplementary data to reflect this change, or written detailed methods. I recommend rejection of this MS, but am confident that the authors can easily make the necessary revisions to make this a paper consistent with the data and without misrepresentations of the data and unsupported claims.

Further comments pertaining to this review of the revised MS are given below in blue.

Reply: We offer our gratitude for the positive comment about our recalculated age. The methods, in fact, has been included in the Methods part titled 'Tuff and zircon dating'. Please see lines 162–183. We used the age with concordance > 98% for analyzing, and ten youngest ages for calculating the weighted mean. This has been already indicated in the main text (lines 59–62). To make it clear, we also added explanation indicating why the older ages were excluded from the calculation (line 62–65). The detailed reply relating to the methods see Comment [5].

[3] Basically, the data is treated better, but the methods aren't described and the presentation remains sloppy in the supplement. These issues need to be corrected before final acceptance.

However, I am confident that you will be able to address them.

Reply: Thanks. Please see comments [2] and [5].

[4] Specific problems that still need to be addressed:

The geochronology data is treated somewhat better. I would have included one more older grain in the date interpretation. This interpretation would be more statistically valid, MSWD = 0.6 rather than MSWD of 0.3. However, doing this does not change the date significantly. So, it does not really matter.

Reply: Thanks. We have added the older age grains in the date interpretation. Please see lines 62–63. We have changed the value of MSWD from 0.3 to 0.6. Please see Line 60, Fig.1, and Supplementary Fig. 5.

[5] The comment that this sample does not contain Pb-loss is difficult to evaluate. Subtle Pb-loss may very well exist. Regardless, Pb-loss will not show up very well on either a Wetherill or a Tera-Wasserburg concordia for rocks of this age. So, the inclusion of this plot does not preclude Pb-loss. Their best argument for this interpretation is that the dates cluster very well.

A more significant problem is that the U-Pb methods are not described in the supplementary material or in the main text (i.e., what machine was used, how were the ratios corrected for instrumental fractionation, what reference materials were used to calibrate corrections, and which reference materials were used to assess external reproducibility?). This information absolutely needs to be in the supplementary material.

Reply: Thanks. The Methods part was not described in the Supplementary Information but has been included in the Methods part titled ‘Tuff and zircon dating’ (lines 162–183). We have made this part more detailed. The machine we used is a Nu Instruments Multiple Collector (MC) ICP-MS with a Resonetics RESOLUTION M-50-HR Excimer Laser Ablation System (lines 169–170). The zircon 91500 was used as an external calibration sample to evaluate the magnitude of mass bias and inter-elemental fractionation (lines 173–175). The standard zircons 91500 and GJ-1 were measured for calibration corrections (lines 172–173). The standard zircon GJ-1 was used to evaluate the accuracy and precision of the laser-ablation results, i.e., the external reproducibility (lines 175–176).

[6] Neither reference material used in this study is appropriate for evaluating reproducibility of $^{206}\text{Pb}/^{238}\text{U}$ dates, as they are both (GJ-1 and 91500) intended to be used as reference materials for $^{207}\text{Pb}/^{206}\text{Pb}$ ratios. The $^{207}\text{Pb}/^{206}\text{Pb}$ dates for zircon reference material 91500 report negative uncertainties, which is not meaningful and must reflect a mistake. This issue needs to be corrected.

Reply: Thank you for your advice. We have corrected the negative uncertainties of 91500 (Please see the Supplementary Table 1), and recalculated the age as 72.1 ± 0.3 . We have also adjusted the data in Figure 1, Supplementary Figure 5, and Supplementary Table 1 accordingly. Also please see lines 23, 32, 60, 64 and 83.

[7] The caption on the geochronology figure in the supplement still mentions the histogram rather than the tera-wasserburg plot and rank order plot. Also, the uncertainties need to be identified in this caption (1 sigma vs. 2 sigma).

Reply: Thank you very much for pointing this out. We have corrected the caption by using ‘Tera-Wasserburg plot’ and ‘rank order plot’, and the 2σ analytical uncertainty was reflected in the Fig.1 and Supplementary Fig. 5. Please see lines 86–88 in the Supplementary Information.

[8] In sum, the data interpretation is better and acceptable and is used to justify a late Campanian age. This age interpretation is more reasonable than the previous interpretation. However, the documentation of the U-Pb geochronology methods and presentation of the data in the supplementary material needs to be improved before publication. In the long term, I recommend that the laboratory doing these analyses procure a zircon reference material appropriate for calibrating $^{206}\text{Pb}/^{238}\text{U}$ ratios or at least one that is appropriate for testing the external reproducibility of this measurement.

Reply: Thanks a lot! Please see comments [2], [5] and [6].

[9] Main text

Line 12 Change ‘Univ’ to ‘University of’.

Line 23 Add ‘Upper Campanian’ before ‘~72.2 Ma’.

Line 23-24 Change ‘generated before the K-Pg extinction event’ to ‘which was generated >6 m.y. before the K-Pg extinction event at ~66 Ma.’

Line 31 Change ‘are’ to ‘were’.

Line 33 Add ‘millions of years’ before ‘prior to’.

Line 33 Change ‘fills’ to ‘fill’.

Line 34, Change ‘Tilin amber biota fills a critical insect fauna gap and providing a rare insight into an end-Cretaceous forest ecosystem.’ to ‘Tilin amber biota fill a critical insect faunal gap and provide rare insight into the late Cretaceous forest ecosystem.’.

Line 36 Change ‘became forest’ to ‘dominated forests’.

Line 37 Change ‘that occurred in’ to ‘at’.

Line 37-39 Delete ‘Both events led to a reorganization of terrestrial ecosystems^{4,5}, and probably strongly influenced the evolution of insects and other terrestrial animals^{6,7}.’.

Line 40 add ‘to date’ after ‘insect faunas’.

Line 41 Add ‘the reorganization of terrestrial ecosystem and’ after the ‘understand of’.

Line 42 Add ‘the evolution of’ before ‘insects’.

Line 50 Change ‘in’ to ‘on’.

Line 56 Change ‘grain’ to ‘grains’.

Line 65 Change ‘its age implication’ to ‘their age implications’.

Line 70-73 Move the following sentence to SM: ‘*Sphenodiscus* sp. has the whorl section stouter than that of *S. lotatus*, and the external suture with the adventive lobes much smaller than the lateral so that the adventive saddles are rather small and has lesser incisions than *S. lotatus*. *Sphenodiscus* sp. resembles *S. ubaghsi*, but differs in having obviously ventrolateral tubercles.

Line 119 Add ‘the’ before insect ‘families’.

Line 121 Add ‘,’ after ‘Diapriidae’.

Line 125-129 Change ‘the biting midge (Ceratopogonidae) genus *Protoculicoides* was known from Albian Spanish, Cenomanian French and Kachin, Turonian New Jersey, Coniacian-Santonian Taimyr, and early/mid-Campanian Canadian ambers²³; and the mantis (Mantodea) genus *Burmantis* only occurred in Barremian Lebanese and Kachin ambers²⁴. These findings

further confirm the Cretaceous age of Tilin amber.’ to ‘the biting midge (Ceratopogonidae) genus *Protoculicoides* was known from ambers¹⁹ in Albian (Spain), Cenomanian (France and Kachin), Turonian (New Jersey), Coniacian-Santonian (Taimyr, Russia), and early/middle Campanian (Canada); in contrast, the mantis (Mantodea) genus *Burmantis* is only known from the Barremian ambers (Lebanese and Kachin)²⁰’.

Line 150 Add ‘had already begun by the late Campanian and more than 6 million years’ before ‘prior to’.

Line 156 Change ‘approximately’ to ‘over’.

Line 157-159 Delete ‘Its proximity to the K-Pg extinction event makes the deposit an important point for comparisons across the event, as well as to previous assemblages within the Cretaceous.’

Line 163 Change ‘and the K-Pg extinction event’ to ‘during the late Campanian’.

Reply: We thank the reviewer for the careful check and detailed suggestions. We have accepted all these suggestions and highlighted the changes with yellow colour in the revised main text.

[10] **Supplementary information**

Line 28 Delete ‘closest to the K/Pg boundary are’

Line 29 Change occurring to occurred

Line 30 Change boundary to KPg

Line 30 Delete ‘and’

Line 31 Add ‘KPg’

Line 38 Change ‘an end-Cretaceous’ to ‘a Late Campanian’

Reply: Thanks. We accepted all these suggestions. The changes were marked in yellow colour in the revised Supplementary Information.

[11] Line 82-57 Supplementary Figure 4: Some more information should be given for this figure to tell the reader what it is and what the significance of these fotos is, where are they stratigraphically? Also Figs a and c are too dark to be useful, label a is invisible.

Reply: Information of these photos have already given in Lines 16–19 of the Supplementary Information. Figures a and c were adjusted to be more bright, and label a was added in this figure. There are no precise stratigraphic position for c–d since they were collected by the local miner, and we just know the relative position.

[12] Line 95-98 Supplementary Figure 5: Basically, the data is treated better, but the methods aren’t described and the presentation remains sloppy in the supplement. These issues need to be corrected before final acceptance. See Review 2nd round for specific comments on what is required.

Reply: Please see comments [2] and [5].

[13] Line 103-106 Supplementary Figure 6: The detailed description of ammonites in the text should be reduced and placed in Supplementary data with the ammonite illustrations. The text by

itself without the illustrations is not too useful or meaningful. The authors should give a short statement in the text and refer to supplementary material.

Reply: Thanks, we have moved the related part to Supplementary Information according to the reviewer's suggestion. In the main text, the reference of this supplementary figure has been given. Please see lines 71–72 in the main text.

[14] Line 111-112 Supplementary Figure 7: There should be explanation with this figure rather than just some figure with virtually no information.

Reply: The explanation of this figure has already added in the main text. Please see lines 92–96. We have also added a short explanation in the caption. Please see lines 95–96 in the Supplementary Information.

REVIEWERS' COMMENTS:

Reviewer #1 (Remarks to the Author):

The authors have now eliminated all claims to provide information on the KPG boundary event based on the late Campanian Tiliin amber biota. I'm pleased with the current version which presents an important amber biota and adds significantly to the current information available on this topic. The authors have satisfactorily revised all misleading interpretations pointed out in the earlier reviews. I'm pleased to accept the current version for publication.

REVIEWERS' COMMENTS:

Reviewer #1 (Remarks to the Author):

The authors have now eliminated all claims to provide information on the KPG boundary event based on the late Campanian Tilin amber biota. I'm pleased with the current version which presents an important amber biota and adds significantly to the current information available on this topic. The authors have satisfactorily revised all misleading interpretations pointed out in the earlier reviews. I'm pleased to accept the current version for publication.

Reply: Thank you very much for your comments to this manuscript.